# Policy formulation and actor roles in the expanded Kenyan free maternity policy (Linda Mama): A policy analysis

Boniface Oyugi[1,2], Zilper Audi-Poquillon[3]*, Sally Kendall[2], Stephen Peckham[2], Edwine Barasa[4,5]

1 M and E Advisory Group, Nairobi, Kenya, 2 Centre for Health Services Studies (CHSS), University of Kent, Canterbury, United Kingdom, 3 Department of Health Policy, London School of Economics and Political Science, London, United Kingdom, 4 Health Economics Research Unit (HERU), KEMRI-Wellcome Trust Research Programme, Nairobi, Kenya, 5 Nuffield Department of Medicine, Center for Tropical Medicine and Global Health, University of Oxford, Oxford, United Kingdom

* z.a.audi-poquillon@lse.ac.uk

**Data Availability Statement:** The authors confirm that the data supporting the findings of this study are available within the article and its supplementary materials. Additionally, the data can

## Abstract

In 2013, Kenya implemented free maternity services, later expanded in 2016 into the 'Linda Mama' policy to provide essential health services for pregnant women. This study explored the policy formulation background, processes, content, and actors' roles in formulation and implementation. Using a convergent parallel mixed-methods case study design, we reviewed documents and conducted in-depth interviews with national stakeholders, county officials, and healthcare workers. We applied a theoretical framework capturing the background and context, processes, content, and actors. The study spanned national, county, and facility levels within Kenya's health system. Data were audio-recorded, transcribed, and analyzed using a framework thematic approach. Findings showed that political imperatives and global and national goals shaped the expanded policy, drawing on previous learnings. Actor power played a crucial role in shaping policy direction, reflecting their interests and capacity to influence decisions. The policy aimed to improve coverage and administrative efficiency, with NHIF becoming the primary purchaser of services to ensure sustainability and address legal challenges. The policy design, marked by conflicts and time pressures, required a collaborative approach to reconcile design and costing differences. Despite differing interests, discussions and dialogues were essential for leadership and conflict management, culminating in key policy documents. A committee was established for stakeholders to freely discuss and debate the policy design, enabling relevant players to devise solutions and fostering joint commitment for implementation. Government officials, development partners, and representatives significantly influenced policy formulation. Beneficiary representatives had limited awareness of public participation opportunities. National and county actors supported achieving audit, research, financing, and strategic operational goals crucial for policy implementation. In conclusion, this study highlights the continued significance of policy analysis frameworks and theories in understanding the complex nature of policy development. These findings offer valuable insights for countries designing or redesigning healthcare policies and provide relevant information to academic communities.

be accessed through the first author's PhD thesis at [11], https://kar.kent.ac.uk/88358/ (It is currently in embargo until February 2025), when all the resulting papers and books have been published).

**Funding:** We gratefully acknowledge the financial support provided by the Commonwealth Scholarship Commission (KECS- 2017- 266), which supported BO's PhD study, on which the data and the results of this study are based, and the Economic and Social Research Council (grant number ES/P000622/1) for covering the publication costs of this study. The funding agencies did not play any role in the analysis, interpretation of the results or manuscript writing.

**Competing interests:** The authors have declared that no competing interests exist.

## Introduction

Kenya has significantly reduced neonatal and mortality rates and improved health service coverage over the past few decades. For example, the maternal mortality ratio (MMR) decreased by approximately 6% (from 564 in 2000 to 530 in 2020), [1] while the neonatal mortality rate reduced from 33 to 21 deaths per 1,000 live births between 1990 and 2022 [2]. However, despite this progress, many women, neonates and children in the country still experience morbidity and mortality from preventable pregnancy and child health-related causes [3]. Over the past two decades, there has been an increase in live births assisted by a skilled provider, from 41% in 2003 to 89% in 2022 [2]. This increase has been attributed to the country's advances in health system reforms that have provided affordable and equitable access to essential health services [4,5]. Some of the reforms go back to the 1980s. For instance, user fees were introduced in public health facilities in the 1980s, and those for outpatient care were suspended in 1990 because of equity concerns but later reintroduced in 1991 [6]. Free maternal delivery was introduced in all public healthcare facilities in 2007, but its formulation and implementation were not well documented [6].

In 2013, following a presidential policy directive, Free Maternity Services (FMS) was introduced in all public health facilities [7,8]. The National Government financed the FMS policy with the funds paid directly to healthcare facilities at Ksh2500 (approximately 27 USDs as of the implementation time) for every delivery in primary healthcare facilities, whereas the sub-county hospitals were reimbursed KSh5000 (approximately 55 USDs) for every delivery, normal or caesarean [4]. However, it faced implementation challenges, specifically poor service delivery, due to inadequate preparation before its rollout [9]. Thus, in October 2016, the National Government unveiled an expanded free maternity policy (FMP) called 'Linda Mama' [Swahili word for—take care of the mother], managed by the National Health Insurance Fund (NHIF) to address the challenges from the previous policy [10,11]. It provided an expanded benefits package of essential maternal services for pregnant women, ensuring that all women could access them based on their needs rather than their ability to pay for them [11]. Linda Mama aimed to achieve universal access to maternal and child health services and contribute to the country's progress towards Universal Health Coverage (UHC).

Since its implementation, several studies have reviewed the Linda Mama programme but have only focused on its implementation, effects and impacts, quality of care and the cost [6,10,12–17]. To our knowledge, no study so far, has documented this policy formulation process, and there remains a dearth of knowledge in this area. Yet, studies have shown how problems with policy execution could be enhanced by having a joint consideration of policy design (sometimes defined as crafting comprehensive causal assumptions, goals and visions, rules, tools, strategies and organisations to address a particular policy problem) and execution. Policymakers create policies that deliver desired results by carefully aligning the problems, solutions, interests, and organisational resources and properly working on the policy design, which postulates plausible scenarios and anticipates future implementation problems [18]. Further, policy formulation and agenda setting are essential fields of enquiry that can give insights into the complex formulation process and show how getting and maintaining policy issues on the agenda is a crucial part of decisions made during policy development and implementation [19]. Most policy development or formulation (whether as an intent, a written document, or a practice) does not often follow a particular format as it is a complex and intertwined process [19,20]. It isn't easy to predict.

Studies have shown why some issues make it onto the policy agenda while others fail. For example, some have argued that the structure of organisations could explain why some issues are considered and have an advantage vis-a-vis the alternatives [21]. At the same time,

individuals' or even institutional processes could influence what is to be addressed at any given time. Others have emphasised the role of external events or public opinions and how these two combine with political incentives to shift attention to the policy agenda [21]. Some issues are likely to get a better response from stakeholders if they have high legitimacy, are highly feasible, and have high support [19,22]. Besides, framing the problem is equally important as it influences how policymakers tackle it. The strength of the organisation and individuals concerned with the issue is noteworthy, as well as how those involved understand and portray it [23]. Finally, the features of the problem (issue characteristics) and the political context (the environment in which the actors operate) also play a role [23].

This study, therefore, aims to advance the understanding of Kenya's expanded FMP agenda setting and the policy formulation process. It provides evidence of the issues that were included in the policy formulation agenda and how they were selected. Additionally, it explores the role of the actors in the expanded FMP process. By achieving these objectives, this study contributes to the knowledge in this area and provides valuable insights for policymakers and researchers.

## Methods

### The guiding conceptual and analytical framework

This policy analysis addresses research questions (broadly categorised under context, processes, content, and actors) using a conceptual framework derived from literature review and health policy analysis frameworks (Table 1) [24]. Each of the areas is defined to fit into this study focus. Firstly, to understand the background and the context in which the expanded FMP was implemented, we defined the policy's objectives, considering political, social, and economic factors at local and national levels that potentially influenced the formulation agenda [19]. We did this through an initial situation analysis that helped to establish the problem's

**Table 1. Guiding framework for evaluating the policy process.**

| Area of analysis | Description | Evidence |
|---|---|---|
| Background and context of the policy | It clarifies the policy's objectives, considering political, social, and economic factors at local and national levels that influence outcomes. It includes an initial situation analysis to establish the problem's scope and importance, and outlines the vision, ownership, and leadership necessary to address actor resistance. | Buse et al. [19] and Hercot et al.'s [25]. |
| Policy formulation processes | How the policy was initiated, (re)formulated or developed, negotiated, communicated, and implemented | Walt and Gilson' s [27] and Buse et al. [19] |
| Content of the policy | These are the policy's objectives, which refer to the implicit contract between the government and the users (the benefits package of the expanded free maternity package) and the contract between the government and the health facilities (the 'resource contract'), which includes the population and services covered and the funding sources. | Hercot et al.'s [25] |
| Roles of the actors in the formulation and the implementation | Important players in the policy process whose roles, power, and influence determine the formulation and implementation of a policy. | Buse et al. [19] |

Source: Author constructed from literature review.

scope and importance and outlined the vision, ownership, and leadership that were key in setting the priorities of the policies [25]. This allowed us to show how a window of opportunity was created to develop the 'inventory phase' of the policy [26]. Secondly, we analysed the policy formulation process, capturing how the policy was initiated, (re)formulated or developed, negotiated, communicated, and implemented [19,27]. Thirdly, we looked at the content of the policy formulation, which covered the policy's objectives. In this study, we define them in two ways: the implicit contract between the government and the users (the benefits package of the expanded free maternity package) and the contract between the government and the health facilities (the 'resource contract'), which includes the population and services covered and the funding sources [25]. Lastly, we analysed actors' roles, power, and influence during the formulation and implementation of the expanded FMP [19]. We did it through a stakeholder analysis approach, which was vital to help understand the intentions, interrelations, agendas, interests, and the actors' influence on decision-making processes [28–31].

## Study design

This study was part of a larger study that examined the policy process, quality and cost of free maternal healthcare in Kenya, based on a convergent parallel mixed-methods case study design [11]. The qualitative data reported in this study involved document review, key informant interviews (KIIs) with national stakeholders, and in-depth interviews with county officials and health care workers (HCWs), which focused on and reports about Kenya's expanded FMP agenda setting and the policy formulation process, and role of actors. Other parts of the study are published elsewhere [5,17,32–34]. The study was conducted between November 2018 and June 2019.

## Study setting

Kenya's governance structure is devolved, with one national government and 47 semi-independent county governments [35]. Under the health sector, the national government is responsible for policymaking and regulation, while the county governments oversee the provision of health services [36]. The health system has a pluralistic approach to providing and financing services [36]. The service provision is divided into six levels. Level 1 includes community units managed by community health workers who provide promotive services. Levels 2 (dispensaries) and 3 (health centers) offer primary healthcare and coordinate community health efforts. Levels 4 and 5 provide curative services as county secondary referral facilities, and some also serve as training centers. Level 6 consists of semi-autonomous tertiary facilities offering specialized care and functioning as training institutions [36]. On financing, Kenya's health sector receives funding from three main sources: the government, households, and donors, which contributed 47.41%, 22.77%, and 18.43% of the total health expenditure in 2021, respectively [37].

This study was conducted across multiple levels within the Kenyan health system: national level, county level, and facility/ hospital level to capture the entirety of the policy right from formulation to implementation. At the national level, the setting included the MoH, the NHIF, and development partner and other agencies involved in formulating the expanded FMP. At the county level, this study was conducted in Kiambu County in Kenya. Kiambu was purposively chosen because it has unique sociodemographic characteristics, health indicators, and population size [38–40]. For instance, it is the second-most populous county in Kenya after Nairobi City County, with a population of 2,417,735 [38], with a good balance between the population living in urban and rural areas [38], thereby providing a unique opportunity to highlight the implementation of the policy in both ends. The county has a higher maternal

mortality trend than other counties around Nairobi in the Central Region [41]. Further, 89.2% of births in the county happen in a health facility, 98.2% of births provided by a skilled provider, 67% of women aged 15–49 who had a live birth had 4+ antenatal visits, and 89% of women aged 15–49 had a postnatal check during the first two days after birth [2]. Considering the study's funding limitations, the researchers purposefully selected a county near Nairobi County that would provide a sufficient number of facilities/recruiters to carry out the assignment. The majority of care facilities in Kiambu are primary care facilities, consisting of 70 tier 2 dispensaries and tier 3 health centers, which have lower volumes compared to the 13 tier 4 hospitals and 1 tier 5 inter-county facility with higher volumes [42]. This was also essential for the facilities to have adequate capacity to recruit the target sample within a short period, obtain study clearance quickly and have easy access to the county. Additionally, Kiambu is one of the counties that successfully implemented an innovative project on the voucher management scheme, targeting indigent mothers, a nearly similar concept to the FMP [43–47].

We purposefully selected three study facilities in the purposively chosen county: a level 3 (considered a low volume–few numbers of clients), a level 4 (medium volume), and a level 5 (high volume). The facilities were chosen in consultation with the county team to provide nuanced, unique sub-counties dynamics given their richness in information and characteristics. (See Table 1).

## Study population and sampling

We sampled the study participants through purposive and snowballing techniques, based on their knowledge of the formulation and implementation of the expanded FMP. The goal was to gain a thorough understanding of the expanded FMP by engaging knowledgeable participants instead of interviewing a representative sample of all stakeholders [48,49]. The population included national level policymakers that were involved in the formulation processes such as Ministry of Health officials, NHIF officials, development partners and civil society (n = 15); subnational level/County policymakers such as the county department of health officials (senior-level and middle-level managers) (n = 3); Public hospital managers (Medical superintendent, facility-level managers, department in-charges, hospital administrator) (n = 9); and Health Workers (nursing officers, clinical officers, accounting/ clerical officers) (n = 9) that were equally involved in the formulation and implementation of the policy. As this study is drawn from a larger study [11], the results from the recipients of the policy–the mothers–are not reported here as they have been published elsewhere [17,33,34]; however, some of the concepts they provided from the larger studies were used in elucidation the roles of actors in formulation and implementation as summarised in Tables 3–5. Further, in this study, we reviewed documents, whose contents were used in addressing the research questions and they included, legal documents (n = 10), websites (n = 10), and other documents (n = 9) as described in Table 2.

## Data collection and analysis

Data collection occurred in three stages. The first stage involved a critical review of documents listed in Table 2 to understand the policy's envisaged design, the roles of actors, and the documented processes during formulation. These documents were sourced from government websites and key informants and the information in them were extracted into MS Word. In the second stage, in-depth key informant interviews (KIIs) were conducted with national-level respondents listed in Table 2 and discussed in detail in the above section. Semi-structured interview guides (S1 Appendix) were used to gather detailed information on the emergence and policy process of the FM policy, focusing on aspects less likely to be found in policy documents, such as the roles and interests of various actors. The third stage involved in-depth

**Table 2. Summary of respondents and documents reviewed.**

|  | Female | Male | Total |
|---|---|---|---|
| **County-level and health facility respondents** |  |  |  |
| The county department of health officials (senior-level and middle-level managers) | 2 | 1 | 3 |
| Public hospital managers (Medical superintendent, facility-level managers, department in-charges, hospital administrator) | 7 | 2 | 9 |
| Health Workers (nursing officers, clinical officers, accounting/ clerical officers) | 6 | 3 | 9 |
| **Total** | **15** | **6** | **21** |
| **National level respondents** |  |  |  |
| Ministry of Health officials | 4 | 1 | 5 |
| NHIF officials | 3 | 0 | 3 |
| Development partners and civil society | 2 | 5 | 7 |
| **Total** | **9** | **6** | **15** |

**Documents Reviewed**

| Category | Item |
|---|---|
| Legal documents | 1. The Constitution of Kenya (Article 187, Article 43)<br>2. The Health Act (Article 2, Article 5, Article 6, Article 7)<br>3. National Hospital Insurance Fund Act<br>4. The County Governments Act, 2012<br>5. Intergovernmental Relations Act, 2012 (section 25)<br>6. Legal Notice 137–183 of August 2013 (which exercise the powers conferred by section 23(1) of the Transition to Devolved Government Act, 2012<br>7. Legal Notice No.34 National Government Regulation<br>8. Kenya Vision 2030<br>9. The Health Sector Strategic and Investment Plan (HSSP) 2019–2023<br>10. Implementation of the Agenda 2030 for sustainable development |
| Websites | 1. The Standard Group PLC<br>2. The Nation Media Group<br>3. Kenya Ministry of Health<br>4. The World Health Organisation<br>5. The World Bank<br>6. Other websites on key actors that captured element of the expanded FMP formulation and implementation (n = 5) |
| Other documents | 1. Linda Mama implementation manual<br>2. NHIF PowerPoint presentations on Linda Mama (n = 2)<br>3. Draft free maternity service policy<br>4. Rapid assessment of Linda Mama report<br>5. Other Linda Mama case studies reports (n = 2)<br>6. Facilities maternal and child health progress charts and wall hangings/ posters<br>7. Published scientific documents on free maternity policy (n = 1) |

interviews with implementers at the county and facility levels also summarised in Table 2. Using semi-structured guides, these interviews aimed to capture the formulation processes and implementation experiences of the FM policy. A laddered approach was employed [50], starting with health facility workers, then facility in-charges, and finally county officials, to minimize biases from power differentials and provide appropriate context by asking specific questions rather than general ones, considering that people base their responses on their every-day routine experiences [51]. Also the questions were designed to progress from descriptive and non-intrusive to more invasive, addressing knowledge gaps and personal beliefs [50]. We first piloted the interview guides (S1 Appendix) in a non-participating facility, to ensure the questions captured all the aspects of our research objective. The interviews were conducted in English and audio-recorded, each lasting between 30–60 minutes.

**Table 3. Actors roles, interest, influences and position on the formulation and implementation process of the expanded FMP.**

| | Category of actors | Role in formulation | Role in implementation | Interest | Level of power | Position |
|---|---|---|---|---|---|---|
| **Elected officials** | The Presidency | + | + | High | High | Supportive |
| | The members of parliament and senate | No | + | Low | Low | Middle support |
| | County Governor | No | +++ | Medium | High | Middle support |
| | Member of county assembly | No | + | High | High | Middle support |
| **Appointed officials/ offices** | Office of the Auditor General | No | + | Low | Low | Middle support |
| | Council of Governors | +++ | +++ | High | High | Supportive |
| | The National Treasury | +++ | +++ | High | High | Supportive |
| | Cabinet secretary, Principal secretary for health, and Director General (National) | +++ | ++ | High | High | Supportive |
| | MoH-Department of policy, planning and health Financing (Division of Health Policy and planning and division of healthcare financing) (National) | + | + | Medium | Medium | Supportive |
| | MoH-Department of preventative and promotive health (Division of Family Health) (National) | + | +++ | High | High | Supportive |
| | MoH-Other departments and divisions (Standards and quality assurance and regulations, M and E) | + | + | Medium | Medium | Middle support |
| | County Executive Committee (CEC)–Health | No | + | Medium | High | Supportive |
| | County Chief officer of health (County) | No | +++ | High | High | Supportive |
| | The summit ((CHMT) County directors of Health, Administration and planning and their deputies) | No | +++ | High | Medium | Supportive |
| | The County Treasury (Includes County accountants) | No | +++ | Medium | Medium | Supportive |
| | County NHIF focal person | No | +++ | High | Low | Supportive |
| **Purchaser of health services** | NHIF (National level) | +++ | +++ | High | High | Supportive |
| | NHIF (County offices) | No | +++ | High | Medium | Supportive |
| **Member of interest groups** | The Church (SUPKEM, Council of churches) | + | + | Medium | High | Supportive |
| | The Kenya Private Sector Alliance | ++ | +++ | High | High | Supportive |
| | HCWs Unions | ++ | +++ | High | High | Supportive |
| **Donors and development partners** | The World Bank | +++ | +++ | High | High | Supportive |
| | WHO | +++ | +++ | High | High | Supportive |
| | JICA | +++ | +++ | High | High | Supportive |
| | UN agencies (UNFPA) | + | ++ | High | Medium | Supportive |
| | AMREF | + | ++ | High | High | Supportive |
| | USAID | +++ | +++ | High | High | Supportive |
| | Marie Stopes International | + | +++ | High | Low | Supportive |
| | Population service International | + | +++ | High | Low | Supportive |
| | PharmAcess | + | +++ | High | Low | Supportive |
| **Civil society** | Kenya Human Rights Commission | + | + | Low | Low | Immobilised |
| | KELiN | + | No | Low | Low | Immobilised |
| | Centre for Reproductive Rights | No | + | Low | Low | Middle support |
| **Beneficiaries** | Individual citizens (Men and women) | + | +++ | High | Low | Supportive |
| | Private health facilities | +++ | +++ | High | High | Supportive |
| | Public health facilities | + | +++ | High | Medium | Supportive |

(*Continued*)

**Table 3.** (Continued)

| | Category of actors | Role in formulation | Role in implementation | Interest | Level of power | Position |
|---|---|---|---|---|---|---|
| **Academia and researchers** | Kemri Wellcome Trust | No | ++ | High | Medium | Supportive |
| | Population Council | No | + | Medium | Low | Middle support |
| | Mannion Daniels and Options Consultancy | No | + | Medium | Low | Middle support |
| | ThinkWell | No | + | High | Low | Middle support |
| **Media** | Local and international media | ++ | +++ | High | High | Supportive |
| **Other** | Beyond Zero | No | +++ | High | High | Supportive |
| | Jacaranda Health | No | +++ | High | Low | Supportive |
| | Philips | No | ++ | High | Low | Supportive |
| | AfyaTu | No | + | High | Low | Supportive |
| *Key*: | +++: Very good involvement; ++: Good involvement; +: Partial or weak involvement; No: no evidence of involvement. | | | | | |

Source: Author, extracted from a review of documents, the IDIs, KIIs, or Exit interviews (Eis) (**Note:** It is plausible that some actors may have been omitted because they were not apparent in the document reviews or the IDIs, KIIs, or EIs. The EIs here have been discussed in detail else [11]).

All the interviews were transcribed verbatim in English and compared against their respective audio files by the first author (BO) for transcription and translation accuracy. All the validated transcripts and extracted information form the document reviews were imported into NVivo 12. The data were analysed thematically [52–54] following the conceptual framework described prior. One researcher (BO) assigned unique identifiers to the data, familiarised himself with the data through immersion and repeatedly read and re-read the transcripts. He then started by developing 'lower-order premises evident in the text' [55] through open coding (assigning codes to portions of data) [56], thereby creating an initial coding framework. Study team members (SK and SP) reviewed and discussed the initial coding framework, and any discrepancies were appropriately reconciled. The final coding framework was applied (by BO) to the data and later charted the data to allow themes to emerge through comparisons and interpretations.

## Ethical consideration

Ethical approval for this study was obtained from the University of Kent, SSPSSR Students Ethics Committee and AMREF Scientific and Ethics Review Unit in Kenya (Ref: AMREF–ESRC P537/2018). Further, we received permission to conduct the study from all the healthcare facilities where the study was conducted and additional clearance to conduct research from the County Government of Kiambu, Department of Health Services (Ref. No: KIAMBU/HRDU/AUTHO/2018/10/31/Oyugi B). We also got written informed consent from the respondents before conducting the interviews, after informing them about the purpose of the study and their right to withdraw consent at any point. They were also assured of their confidentiality, and that their data would be reported in an aggregated format, and anonymised to protect their identities, throughout the course of this study.

## Results

This section presents results in four main areas: a) the background priorities and the context in which the expanded FMP was developed and executed; b) policy formulation processes, detailing how the policy was initiated, reformulated, negotiated, communicated, and implemented; c) policy content, defining the benefits package for users and the resource contract

**Table 4. Force field analysis map showing the level of influence and power of actors.**

| Level of power | Proponent | | | Opponents | | | |
|---|---|---|---|---|---|---|---|
| | High support | Middle | Low | Non mobilised | Low | Middle | High opposition |
| High | The Presidency | County Governor | | | | | |
| | Council of Governors | Member of county assembly | | | | | |
| | The National Treasury | | | | | | |
| | Cabinet secretary, Principal secretary for health, and Director General (National) | | | | | | |
| | MoH-Department of preventative and promotive health (Division of Family Health) (National) | | | | | | |
| | County Executive Committee (CEC)–Health | | | | | | |
| | County Chief officer of health (County) | | | | | | |
| | NHIF (National level) | | | | | | |
| | The Church (SUPKEM, Council of churches) | | | | | | |
| | The Kenya Private Sector Alliance | | | | | | |
| | HCWs Unions | | | | | | |
| | The World Bank | | | | | | |
| | WHO | | | | | | |
| | JICA | | | | | | |
| | AMREF | | | | | | |
| | USAID | | | | | | |
| | Private health facilities | | | | | | |
| | Local and international media | | | | | | |
| | Beyond Zero | | | | | | |
| Medium | MoH-Department of policy, planning and health Financing (Division of Health Policy and planning and division of healthcare financing) (National) | MoH-Other departments and divisions (Standards and quality assurance and regulations, M and E) | | | | | |
| | The summit ((CHMT) County directors of Health, Administration and planning and their deputies) | | | | | | |
| | The County Treasury (Includes County accountants) | | | | | | |
| | NHIF (County offices) | | | | | | |
| | UN agencies (UNFPA) | | | | | | |
| | Public health facilities | | | | | | |
| | Kemri Wellcome Trust | | | | | | |
| Low | County NHIF focal person | The members of parliament and senators | | Kenya Human Rights Commission | | | |
| | Marie Stopes International | Office of the Auditor General | | KELiN | | | |
| | Population service International | Centre for Reproductive Rights | | | | | |
| | Individual citizens (Men and women) | Population Council | | | | | |
| | Jacaranda Health | Mannion Daniels and Options Consultancy | | | | | |
| | Philips | ThinkWell | | | | | |
| | CHS | | | | | | |

Author, extracted from a review of documents, the IDIs, KIIs, or Exit interviews (Eis) (**Note:** It is plausible that some actors may have been omitted because they were not apparent in the document reviews or the IDIs, KIIs, or EIs. The EIs here have been discussed in detail else [11].

**Table 5. Role of the actors.**

| | Category of actors | Role in formulation | Role in implementation |
|---|---|---|---|
| **Elected officials** | The Presidency | Outlining the *Jubilee Agenda* | Outlining the *Jubilee Agenda*. |
| | The members of parliament and senators | No | • Approving the government spending on Linda mama. |
| | County Governor | No | • Supervision of the CHMT for service provision and financial allocation.<br>• Working in collaboration with other similar projects that are targeted at achieving UHC. |
| | Member of county assembly | No | • Working with the pregnant mothers who provide feedback about the services received. |
| **Appointed officials/ offices** | Office of the Auditor General | No | • Statutory audit of FM policy reports. |
| | Council of Governors | • Supporting initial technical design (technical capacity)<br>• Modalities of implementation at the county level | • Collaborating with the counties to form council of health ministers from the counties to ensure efficient implementation of the policy at the county level. |
| | The National Treasury | • Resource and budgetary costing | • Resource and budgetary allocation. |
| | Cabinet secretary, Principal secretary for health, and Director General (National) | • Oversight of the discussion and direction<br>• Overseeing the implementation of the previous Linda mama Services and transition from the previous FMS to the current Linda Mama | • Providing funds to the NHIF.<br>• Source for funds from the National Treasury and provides strategic, future policy direction in line with the presidential directive of UHC. |
| | MoH-Department of policy, planning and health Financing (Division of Health Policy and planning and division of healthcare financing) (National) | • Limited involvement except advisory | • Limited involvement.<br>• Advisory on Health financing strategies not linked to Linda mama. |
| | MoH-Department of preventative and promotive health (Division of Family Health) (National) | • Limited involvement except advisory | • Providing the overall oversight of the implementation of Linda mama (Providing the technical lead on behalf of MoH).<br>• Monitoring and evaluation of the progress of implementation of UHC for which Linda mama is part. |
| | MoH-Other departments and divisions (Standards and quality assurance and regulations, M and E) | • Limited involvement except advisory | • Limited involvement except advisory. |
| | County Executive Committee (CEC)–Health | No | • Coordinates health services at the county. |
| | County Chief officer of health (County) | No | • Hands on in overseeing the implementation of Linda mama at the county level. |
| | The summit ((CHMT) County directors of Health, Administration and planning and their deputies) | No | • Supervision of the policy outcome.<br>• Providing continuity of supplies and supporting the referral system.<br>• Communication of the policy to the healthcare workers<br>• Employment of the clerks and supervising them. |
| | The County Treasury (Includes County accountants) | No | • Providing approvals to the facilities to spend the cash.<br>• Accountant oversees financial operations. |
| | County NHIF focal person | No | • County NHIF point person who streamlining the hospital accounts and making sure they do the right things.<br>• Overseeing the UHC project for which the Linda mama is part.<br>• Linking with the Beyond zero project to ensure free camps maternal camps are carried out. |
| | Hospital employees (HRIO, NHIF clerk, In charges, Administrators, Other HCWs) | No | • Provide services to the Clients and supporting in their registration. |
| **Purchaser of health services** | NHIF (National level) | • Supporting initial technical design (technical capacity)<br>• Came up with ways of improving coverage (issuing cards and setting up offices in the hospital) | • Overall management of Linda mama.<br>• Creating demand and providing awareness / educating the mothers.<br>• Registration of the members and providing the services. |
| | NHIF (County offices) | No | • Batching of claims form all hospitals in the county. |

*(Continued)*

**Table 5.** (Continued)

| | Category of actors | Role in formulation | Role in implementation |
|---|---|---|---|
| **Member of interest groups** | The Church (SUPKEM, Council of churches | • Provide support on the implementation strategy and the duality of it.<br>• Provide input from members | • Educating the congregations on FM policy. |
| | The Kenya Private Sector Alliance | • Provide support on the implementation strategy and the duality of it.<br>• Provide input from members | • Provide support on the implementation strategy and the duality of it. |
| | HCWs Unions | • Provide support on the implementation strategy and the duality of it.<br>• Provide input from members | • Critiquing the government's implementation process. |
| **Donors and development partners** | The World Bank | • Funding the initial initiative<br>• Supporting initial technical design (technical capacity)<br>• Part of the technical working group discussing the movement | • Participating in the discussions around health reforms in Kenya for which Linda mama is part. |
| | WHO | • Supporting initial technical design (technical capacity) | • Evaluate the legal access rights to health care through independent consultants. |
| | JICA | • Supporting initial technical design (technical capacity) | • Fostering partnerships for UHC. |
| | UN agencies (UNFPA) | • Advocating for inclusion of a broad spectrum of services | • Supporting the MoH to develop the policy and plans and documents. |
| | AMREF | (+) | • Engaging the extensive network of community health volunteers and beyond zero to register mothers in the program. |
| | USAID | • Supporting initial technical design (technical capacity)<br>• Transitioning from the FMS to Linda mama<br>• Supporting the launch of Linda mama through a report | • Directly working with the facilities to enhance the QoC, investing in human resource, investing in supplies and commodities.<br>• Supporting in development of the policies, more so health financing policies.<br>• Working with counties to improve their efficiency in utilisation of the available resources and other resource allocation (PFM act).<br>• Advocacy for increasing resources.<br>• Supporting the District Health Information System (DHIS), and data quality assurance (DQS) in hospitals. |
| | DANIDA | | • Providing equitable fund to improve facilities |
| | Marie Stopes International | (+) | • Capacity building of the provider level for both private and public providers on claim process, accreditation, process ff contracting.<br>• Demand creation by creating awareness of the policy to the community.<br>• Support the government in achieving UHC. |
| | Population Service International | • Discussion with the NHIF on the importance of working with the private sector in informal settlements (more so small and middle-level health facilities) | • Through AHME, working with NHIF to package benefit for the informal sector.<br>• Capacity building/ professional competency/ continuous medical education of the providers.<br>• Monitoring and evaluation/ supervision to ensure quality is adhered.<br>• Demand creation by creating awareness of the policy to the community.<br>• Ensuring that the registered facilities are properly licenced by the professional bodies such as NCK, Clinical officers board, KMPDB.<br>• Conduct their own quality checks in the facility before empanelling to ensuring hospitals have beds, referral equipment; and safecare program for 6–12 months before empanelling so accreditation is guaranteed. |

(*Continued*)

**Table 5.** (Continued)

| | Category of actors | Role in formulation | Role in implementation |
|---|---|---|---|
| **Civil society** | Kenya National Commission on Human Rights | • A review of implementation of programs including FM policy | • A review of implementation of programs including FM policy. |
| | KELiN | No | • Providing legal critique of hinging Linda Mama under NHIF. |
| | Centre for Reproductive Rights | No | • Documenting abuse and disrespect in maternal health setting. |
| **Beneficiaries** | Individual citizens (Men and women) | • Involvement of the community in forums and at the launch | • Registering for the service (self-registration or HCW supported). • Benefiting/utilising the services. |
| | Private health facilities | • Discussion about reimbursement strategies and rates | • Provision of the service to the beneficiaries • Reporting the outcomes. |
| | Public health facilities | • Providing feedback from the previous FMS | • Provision of the service to the beneficiaries • Reporting the outcomes. |
| **Academia and researchers** | Kemri Wellcome Trust | No | • Working with ThinkWell and NHIF to conduct process evaluation of Linda mama. |
| | Population Council | No | • Impact evaluation of removal of fee for FMP on UHC. |
| | Mannion Daniels and Options Consultancy | No | • Evaluating a case study of implementing Linda Mama in Kenya Bungoma County. |
| | ThinkWell | No | • Working with Kemri Wellcome Trust and NHIF to conduct process evaluation of Linda mama. |
| **Media** | Local and international media | No | • Participate in media coverage of progress and critiquing the government where there is no progress. |
| **Other** | Beyond Zero | No | • Engaging the county governments and the NHIF to do a mobile clinic campaign encouraging mothers to register with NHIF and access Linda mama. • Work with like-minded programs and organisation to support maternal care. |
| | Jacaranda Health | No | • Coordinating with the healthcare facilities to conduct health care education and training nurses on the care for patients. • Evaluating satisfaction of client on the services provided. |
| | Philips | No | • Develop innovation and digital solutions for Maternal and Child Health such as Digital labour and delivery solution (DLDS) and Mobile Obstetric Monitoring (MOM). |
| | CHS | No | • Employing PMTCT nurse in maternity. |
| | Aphia Plus | No | • Training stuff on provision of quality care; providing equipment and supplies for maternal care. |
| **KEY:** | **(+):** there is participation, but the interviewees could not reveal; **(-):** there is participation from document review but not outrightly stated;**?:** In depth interviews and document review could not reveal any evidence of the role **Abbreviations:** CHS–Centre for Health Solutions; PMTCT–Prevention of mother to child transmission; NHIF–National Health Insurance Fund; UHC– Universal Health Coverage; FMP–Free Maternity Policy; HCW–Healthcare workers | | |

Author, extracted from a review of documents, the IDIs, KIIs, or Exit interviews (Eis) (**Note:** It is plausible that some actors may have been omitted because they were not apparent in the document reviews or the IDIs, KIIs, or EIs. The EIs here have been discussed in detail else [11]).

with health facilities, covering populations, services, and funding sources; and d) roles of actors in both formulation and implementation.

## Background and the context of the expanded FMP

**Aligning the objectives of the policy with country's legal and policy-guiding instruments.** Based on the document reviews, the expanded policy aimed to align and enhance the

provision of reproductive, maternal, neonatal, and child health (RMNCH) services in line with the Constitution of Kenya, 2010, Kenya's Vision 2030 framework, and the Health Sector Strategic and Investment Plan (HSSP) 2019–2023 [57]. For example, borrowing from the constitution, every citizen is granted the right to quality health and life (including maternal and neonatal care) [35]. At the same time, the social pillar of the Kenya Vision 2030 envisioned a country-wide scale-up of high-impact community health interventions (such as free maternity service) that focused on boosting nutrition, family planning, immunization, sanitation, and safe motherhood [58]. Additionally, the HSSP aimed to increase equitable access to quality services and sustain demand for improved preventive and promotive healthcare [59]. All these were aligning with the objective of the expanded FMP.

**Need to achieve and align with the global agenda.** *Achieving the Sustainable Development Goals (SDGs).* Both document reviews and survey respondents indicated that the expanded FMP was seen as a crucial cornerstone for advancing the SDGs 3.1, 3.2, and 3.7, which focus on reducing maternal mortality, ending preventable deaths of newborns and children under 5, and ensuring universal access to sexual and reproductive health-care services. For instance, the documents on the implementation of the Agenda 2030 for sustainable development, showed that despite some progress in achieving the Millennium Development Goals (MDGs), the country had missed those targets and therefore the FMP was one of the strategies positioned to meet the gaps [60,61]. Similar views were expressed by the study respondents, emphasizing the policy's aim to address the gaps in the MDGs and achieve the maternal and child health goals in the SDGs:

> '. . . previously we had been given the MDGs [Millennium Development Goals], but they didn't work for the 15 years. So now we are working on the SDGs [Sustainable Development Goals]. So being a healthcare worker we are trying our best to ensure that the country and our county achieves its objective [through Linda Mama]'–**(R003, Nursing Officer)**.

*Achieving UHC.* Both document reviews and survey respondents indicated that the expanded FMP was an incentive to boost the quality of maternal care by enhancing access and improving skilled birth attendants (SBA). The respondents reiterated the findings of the Kenya Service Availability and Readiness Assessment survey [62] that had showed the existence of sub-standard care in facilities, and therefore, the enhanced policy was meant to boost maternal and neonatal health, that had deteriorated in quality within the facilities. In addition, there was the need to eliminate financial barriers which would enhance access to free maternity services by pregnant women. As such the respondents perceived that the policy would address concerns of other barriers of access; such as geographical barriers (ensuring that the marginalised populations and those in rural areas have equal access); socio-cultural barriers (ensuring that the use of traditional birth attendants (TBAs) reduces); and service access (ensuring access to services such as testing at ANCs which were previously unavailable), thereby enhancing the achievement of UHC.

> 'Linda Mama also informs Universal Health Coverage, because Universal Health Coverage seeks to enhance, seeks to embrace access to health care providers.'–**(R034, NHIF Officer)**.

As part of enhancing UHC and increasing coverage and access, there was the need to rope in the private sector and faith-based organisations (FBOs), who were not implementers in the previous FM policy. The move was aimed at decongesting public facilities and giving pregnant women more choice. Further, the move was thought of as being imperative in improving access in areas, especially arid and semi-arid areas, where there were not many government

facilities, but that had many FBOs offering services. Eventually this would end up improving efficiency:

*'. . ..we weren't including private, and as you know private almost takes care of 40% of our population, only 60% more often that uses GOK [Government of Kenya]. . .there was a feeling that we are leaving a few people behind, especially in Nairobi, where we have more private facilities than public. . ..so that was one of the driving forces, FBOs [Faith Based Organisation] also thought left behind, who are our partners in so many ways. . ..There was also the consideration of the far front [marginalised] areas where there are private and mostly even FBOs and there is no GOK facility around, but there is a lot of FBOs especially in Turkana, so it was thought [important] for us to improve on access and efficiency'–(R032, MoH Official).*

**Fulfilling a political campaign agenda.** Some respondents perceived the expanded policy to be a political tool used by the government to fulfil a campaign agenda, as captured in the president's *Jubilee Party 2017 Presidential campaign Manifesto [63].* On the other hand, some of the respondents highlighted that the goal of the policy was to fulfil part of the Big Four agenda[64] (the four national agendas that focused on enhancing manufacturing, achieving universal healthcare, ensuring affordable housing, and improving food security in Kenya) that the then president formulated to align his election promises, and policy makers and implementers had no option but to implement it:

*'it was used as a political tool. . .to be able to achieve and acquire power. . .I can tell you in terms of even conceptualizing the idea implementation, politics played its part, and I think the president announced it in 2013 during, is it Madaraka day [Kenyan public holiday]. . .so there is always politics behind some of these things. . .I think it was politically appearing so usually the health workers have no. . .choice but to actually actualize'–(R026, Development partner)*

**Addressing gaps in the previous policy.** *Creating uniformity in service coverage.* It was perceived that the policy, as implemented, needed to have uniformity of service provision across the counties (especially those that had launched their own unique service delivery platforms for their citizens). This would allow citizens from a different county to access services in another county even if they hadn't registered in them. It would enhance competition amongst counties and reduce the incentive to seek healthcare services in counties that had invested more in their healthcare services:

*So, I guess the structure levelled out the advantage that already developed counties would have in terms of service delivery to their citizens. Because all we need is a card or a registration, it doesn't matter which county you came from, and you will be able to access your care from whichever county you go. That is very different from the roll-out you see in other counties, where a county is rolling out its own program and you can only get that care if you are from [reside? in] that county and are registered as a member of that county. That discrimination also ruled out the advantage of having citizens access care in neighbouring [counties] and not pay for it [as] somebody else pays for it.'–(R010, Facility manager).*

*Covering the loss of funding that was previously charged to the mothers for the services, but that would no longer be available.* The facilities needed to continue sustaining the health facilities costs. However, it was also linked to the need to improve the quality of care from the

reimbursements to the facilities (incentivise the facilities). It was noted that the funds would help purchase equipment that the government had been unable to provide, motivate staff through incentives, and employ additional staff:

> *'You know cost sharing; in the past you know, maternity used to generate cash and now it's zero because it's free and services have to run...I think it's purely because of financial reasons'–(R004, Nursing Officer).*

*Rectifying implementation shortcomings of the previous FMP implemented in 2013.* The implementation challenges experienced in the previous FM policy were perceived as the substantial reason for the shift to the expanded FMP. The respondents at both the national and the county level referred to the challenges that had been noted by Tama et al. [9] in their process evaluation as: *lack of exhaustive service package* due to inadequate costing of the services; *data problems*, where facilities were using overstated/ exaggerated utilisation numbers/ figures, rather than mothers' unique identifiers, to get claims from the MoH that were unverifiable; *poor quality of care*, since patients was pushed to the public hospitals that were ill-prepared to tackle the high number of mothers because of inadequate infrastructure. Additionally, there was a *lack of or inadequate communication* of the policy at the grassroot level, which led to poor clarity of the content of the policy, and *disappointed and dissatisfied clients* with the services. The respondents noted that the work was overwhelming to the MoH Department of policy and planning, and reproductive health, which did not have the capacity to manage both the payments and the services:

> *'Data fraud, issues about data verification, data validation to be able to monitor the utilization rate, second the issue of disbursement of money, proper disbursement of money from the Ministry of Health to the health facilities, thirdly is the fact that private sector was left out.'–(R029, Development Partner)*

### Expanded FMP formulation processes

**Private sector's key role in policy formulation.**  The private sector network had a significant interest in the policy formulation phase. The justification of the inclusion of the private sector by the private sector networks and negotiation on the pricing characterised the discussions at the formulation stage. For instance, in using previously conducted research, the networks observed that the developed systems in the private sector would ease the previous FMP (implemented in 2013) implementation challenges in the public sector. As such, the networks opted to push the private sector agenda of their inclusion through the national and county leaderships (using the success of the programs they had implemented prior, such as leveraging their network of community health volunteers which would make them favourable when the policy was finally rolled out). Besides, through their established interests and strengths, as soon as the policy had been formulated, they worked with their networks of private health facilities for advanced accreditation using internal quality of health standards and guidelines before the actual Linda Mama accreditation. They also communicated the packages of the policy in advance to their network of hospitals bypassing the national political policy process to prepare them for accreditation and engagement:

> *'...we gave our input at the National level, but also, we realized that we needed to do some groundwork at the County level, because these are two independent Governments...so at the MOH level, so we went out to the CHMT's [County Health Management Team] at the*

*community level and just made them understand what we are trying to do and why this is important and why they should support us, for them to be able to deliver on free maternity care through the private sector. And so, what we saw the CHMT's do is that they attached themselves to our teams and they did routine supportive supervision with us on a sampling basis just to ensure that what we were telling them was what was on the ground'–**(R030, Development partner).***

The perspective from the private sector network was that the NHIF could not manage (lead) the extensive quality of service that the private sector, the NGO and the FBOs had achieved so far because their reimbursements were perceived as unattractive. The NGO and FBOs were keen to join the policy scheme but proposed that the government raise the reimbursement cost. The concern was that their investment in infrastructure and other additional costs, such as rent and staff, was high:

*'Within our. . .400 plus providers. . ..NHIF cannot manage that quality. . .you see there are so many things, it's [the policy] creating access, and it's creating equity in term of it reaching the poor. . .not so many people appreciate it because it's not attractive in terms of profit but there are a lot of private sectors that are actually improving on that. . . we are not saying you put it at 10000/ = [USD 95.9], we are saying put it at 6000/ = [USD 57.5] there will be demand and they will increase. And then contract other facilities to support you, contract other quality organizations like PSI [Population Service International], MSI [Marie Stopes International] who are known for quality in terms of SRH [sexual reproductive health] to support the Government in terms of quality.'–**(R029, Development partner)**.*

**Costing the policy.**    *The policy was costed based on projected maternal indicators and confirmed funding.* As noted by the respondents, the policy was conceived as an insurance package allowing mothers to use the registration card, guarantee maternal care for up to a year after birth and transition to an NHIF card for outpatient and inpatient services at their facilities of choice. The three bases of costing at the formulation were that a) health is a free good, b) the need for introducing sustainability in developing the provision of maternal care, and c) the expected number of deliveries given the demographic characteristics of the mothers. It was estimated that there would be almost 1 to 1.2 million mothers delivering in the country every year, with a conservative estimate of 25% being insured by entities other than NHIF. Eventually, the FM policy would reach 700,000 mothers by the first year of roll-up, who, after having utilised the services and seen the value of it, would be attracted to it as some form of insurance. Consequently, it was further envisaged that 25% of those who use the free policy services, especially those in quintiles 1 and 2 (a wealth quintile divides a population into five equal parts based on wealth, using indicators like household assets, income, and living standards, to analyze and target socio-economic and health interventions, with Quintile 1 representing the poorest and Quintile 5 the richest [65]), could transition to full insurance after the expiry of *Linda Mama* and start paying for it. The formulation idea was that–in the following years of the policy–with the mothers in quintiles 1,2, and 3 paying a monthly contribution fee to the NHIF for themselves, it would ease pressure on the treasury fund meant for the policy. Additionally, there was a projection of the prospective number of both vaginal and caesarean deliveries that was envisaged, with the assumption that 15–20% of the deliveries would complicate:

*'So, all those were projected, the caesarean section how many do were expected, close to 10%, 15%, the normal ones. . .'–**(R032, MoH Official)** and '. . .when the fund was being put in*

*place, it was assumed that at least fifteen to twenty percent would complicate [result to a complication]. So, they will be catered for by everything else.'–(R024, MoH Official).*

The overall projection cost of the policy was estimated at KES 6.5 billion (USD 62.3 million) (Exchange rate, 1 USD = KES 104.32, which was the rate as of 1st January 2017 at the initiation of the first phase of the implementation of policy (from https://www1.oanda.com/currency/converter/), but the national treasury allocated the MoH KES 4.2 billion (USD 43.1 million) which the respondent deemed sufficient to meet the need.

Further, the funding source for the policy was ascertained. Unlike the previous policy, co-funded by the development partners (JICA and The World Bank), the *Linda Mama* policy was mainly conceived as tax-funded by the national government. However, the respondents noted that there was no specifically earmarked funding for the policy.

*'It is largely funded by our taxes. . .and they are not earmarked, so that means that they could easily be rerouted somewhere else.'–(R023, MoH Official)*

*Concerns over lack of consultation in policy costing.* There was general perception that the costing had not been done in a consultative manner as should be the norm. While the majority of the national-level actors perceived [argued/felt] that the policy had been sufficiently costed including assessing its sustainability, others felt that the program had been arbitrarily costed and had made many costly assumptions:

*'. . .we cost it according to what we spend and then we put an arbitrary amount for miscellaneous, we are not thinking of what is the cost-benefit of this like can you relate every cost you put in, like for a hundred bob we put into healthcare, how many maternal deaths are averted. . .we don't know why; we cannot justify why we put a mark of four billion at least scientifically'–(R023, MoH Official)*

Consequently, the respondents at the meso level felt that the costing was somewhat dubious based on the implementation experience that had left them with more costs to absorb from the service provision:

*'You don't know how the figures of reimbursement were arrived at and noting that every county is a market in its own way, with its own influencers of supply, demand and price. Giving one cut line of a price disadvantages those counties where services are provided at a higher cost compared to counties where services are provided at a lower cost. So, for you to remain in the system you had to absorb some cost and absorbing the cost [would mean] you have a loss. And again, with coming to be like a revenue loss for the county to keep on reimbursing services for which their people are asking for but for which the money funding is not adequate per case.'–(R010, Facility incharge)*

**NHIF chosen as the ultimate purchaser of services.** At the formulation stage, the NHIF was the agency proposed to run the FM fund, as it had the technical and institutional capacity to manage such funds based on its experience. There was no other consideration of transitioning the policy to other organisations other than the NHIF. The majority of the respondents believed that NHIF was undergoing reforms at the time aimed at transitioning to UHC; hence, had better accountability structures. Additionally, NHIF had existing structures and networks of facilities across the country that could be leveraged instead of creating a different program. Besides, NHIF had already been working with the private sector, hence, it was easy to attract

and enroll many facilities. The plan, therefore, was to utilise NHIF instruments to track elements such as average length of stay, quality of care, access, fraud, complaint system and payment of providers:

> 'No there were no other options. . .[be]cause [with] Linda mama. . .we wanted really to involve the low-cost privates to improve the access to mothers which we would not have done as MOH [be]cause MOH can't pay money to private but the law allows NHIF to pay to do that. So that was one of the driving force[s]. NHIF can pay private I mean they have contracts with them, so it was easy we weren't going to start something new'–**(R032, MoH Official).**

While some respondents felt that there were no legal hurdles in working with the NHIF–a semi-autonomous agency under the MoH–document review revealed that civil society organisations had been pushing the narrative of the illegality of the process (see, KELIN Kenya [66]). Additionally, engaging the NHIF would remove the challenge of returning unconsumed money to the treasury as was the norm when the MoH managed the program during the previous policy. The process would allow for exhaustive use of the finance allocated for the project in the rolling years:

> 'So, what happens when the money is within the ministry of health, when the financial year comes to the end, that money goes back to national treasury, through the process, the government budgeting process and it is availed again the following financial year. So in such a scenario where now it goes to NHIF such a body corporate, that money does not have to go back and perhaps if there are pending reimbursements like for instance if I am doing reimbursements for the last quarter of the financial year, April, May, June, and you see like for end of June or end of May, the facilities have to report, then it is compiled, then it is paid, which means it will be paid post, the financial year is ended and the money will already have gotten back. So, they won't be paying for that period. So that kind of challenges they required an institution that can be able to handle that'–**(R024, MoH Official)**

**Development of implementation guidelines and manuals.** The implementation manuals and guidelines were agreed upon at the formulation stage but there were gaps in the monitoring the resulting quality of maternal care from the policy. The formulation committee developed a policy document and a concept note which were eventually taken to the Cabinet for approval. In addition, an implementation manual and a communication strategy were also developed.

To kickstart the process of implementation, a memorandum of understanding (MoU) between the MoH and the NHIF was signed. Correspondingly, financial guidelines were also developed but it was envisaged that the HCWs would rely on the clinical and service provision guidelines that were in use prior to the policy. Notwithstanding that the MoH, through the department of quality standards developed the Kenya Quality Model of Health (KQMH) to provide structured support to counties for general QoC, there was no other guideline on quality maternal care developed at the formulation of the policy, as noted by respondents. There was a feeling that the little attention was being paid to quality guidelines but more on the implementation. The private organisations and development partners proposed to fill up the gap in quality through their own project such as *safe care* which trains and teaches the staff in the facilities on improvement of quality:

> 'MOU with the ministry whereby we looked at how to bring in a bit of technology to assist in the registration of women so that through NHIF system we are able to register first, once you

*register, the next step is that there has to be a confirmation of pregnancy. So, we signed an MOU on what will be our roles in line with the implementation of this product'*–**(R025, NHIF Official)**

*'. . .not squarely on the government, but thanks to development partners and private entities. . .there are very clear projects or approaches that come into address some of these gaps that are identified. So, I say the facilities are willing, but it only works where they have development partners'*–**(R035, Development partner)**

## The content of the policy

The envisaged design envisioned inclusive benefit packages, more infrastructure, and human resources. The envisioned package cuts across maternal care from ANC, delivery, to PNC care, complications and referral services. Furthermore, it was envisaged to take care of the infant within the one-year period in the program (see Fig 1). Further, at the formulation, it was projected that the workload would increase, requiring more investment in infrastructure and human resource:

*'We did not go into those details but we also said as a, the government need to invest on human resources, we are anticipating some increases in human resources, I mean increases in*

| Services for all pregnant women and newborns, for a period of one year | |
|---|---|
| Antenatal care (ANC) | -ANC Profile including Hemoglobin levels, Blood group, Rhesus, Serology, screening for tuberculosis, HIV counseling and testing and urinalysis<br>-Preventive services including tetanus toxoid, intermittent preventive treatment for malaria, deworming, iron and folate supplementation<br>-Prevention of Mother to Child Transmission of HIV (PMTCT) |
| Delivery | -Skilled delivery (including caesarean section) in public facilities and accredited not-for-profit and for-profit private health institutions.<br>-Neonatal care including costs related to pre-term births |
| Postnatal care (PNC) | -Within 48 hours after birth: Analsegics, vitamin A, iron and folate supplemetns, long lasting instecide nets, family planning, PMTCT for HIV positive mothers, treatment or refer any complications for mother, and care for newborn (tretracycline eye ointment, Vitamin K, immunization and birth polio, Infact prophylaxis for HIV if indicated, treat or refer any complications).<br>-Within 1-2 weeks after birth (mother and baby): Screening for cervical cancer, sexually transmitted infections and tuberculosis; and treatment/preventive measures if not previously adminstered<br>-Within 4-6 weeks after birth: Family planning services, screening for cervical cancer, STIs and tuberculosis among others; and immunization as per schedule and early infant diagnosis of HIV<br>-Within 4-6 months after birth: Family planning services, screening for cervical cancer, STI and tuberculosis among others; and immunization as per schedule and vitamin A supplementation |
| Emergency referrals | -Ambulance service |
| Conditions and complications during pregnancy | -Outpatient treatment in in accreditted public, faith-based and selected low-cost private-for-profit facilities<br>-Inpatient treatment in accreditted public, faith-based and selected low cost private-for-profit facilities |
| Children under 1 year | |
| Care for the infant* | -Outpatient services including  treatmet and child welfare clinics  in accreditted public, faith-based and selected low-cost private-for-profit facilities<br>-Inpatient services in accreditted public, faith-based and selected low cost private-for-profit facilities |

*Care for the infant is within the one year period in the programme.

**Fig 1. Benefit packages for Linda mama.** (Source: Adopted from Implementation manual for program managers [67]).

| FACILITY TYPE | SERVICES | FACILITY LEVEL | | |
|---|---|---|---|---|
| | | Level II and III | Level IV and V | Level VI |
| Government healthcare facilities | *Normal delivery* | KES 2,500 | KES 5,000 | KES 6,000 |
| | *Caesarean section* | N/A | KES 5,000 | KES 17,000 |
| | *Antenatal care* | KES 600 for 1st visit then KES 300 for each of the *3* subsequent visits | KES 1,000 for 1st visit then KES 300 for each of the *3* subsequent visits | KES 1,000 for 1st visit then KES 500 for each of the *3* subsequent visits |
| | *Post Natal care* | KES 250 per visit | KES 250 per visit | KES 250 per visit |
| | *Ambulatory services* | Transport for emergency referrals available at a fixed rate based on the number of services | | |
| Private and faith-based facilities | *Normal delivery* | KES 3,500 | KES 6,000 | KES 6,000 |
| | *Caesarean section* | N/A | KES 17,000 | KES 17,000 |
| | *Antenatal care* | KES 1,000 for 1st visit then KES 500 for each of the *3* subsequent visits | KES 1,000 for 1st visit then KES 500 for each of the *3* subsequent visits | KES 1,000 for 1st visit then KES 500 for each of the *3* subsequent visits |
| | *Post Natal care* | KES 250 per visit | KES 250 per visit | KES 250 per visit |
| | *Ambulatory services* | Transport for emergency referrals available at a fixed rate based on the number of services | | |
| KEY: | | OUTPATIENT CARE | INPATIENT CARE | AMBULANCE SERVICES |
| NOTES: | | 1. N/A – Not Applicable<br>2. Exchange rate *1 USD = KES 104.32*; as of *Jan 1, 2017* (https://www1.oanda.com/currency/converter/)<br>3. *Outpatient care:* Facilities are reimbursed based on the number of beneficiaries they provide care. The amounts are risk adjusted<br>4. *Deliveries (either CS /normal):* Facilities are reimbursed as a delivery package where delivery is treated as an event.<br>5. *Inpatient care:* Facilities are reimbursed for additional inpatient services other than deliveries as a rebate (based on the number of days the patients have taken in the hospital) | | |

**Fig 2. Reimbursement rates (Source: Adopted from implementation manual for program managers [67]).**

*workload, I think that should be followed by investment in human resources, investments in commodities and also even infrastructure, other infrastructures, renovation of the maternity wards, those were things that we had anticipated and we recommended investments in those areas as to whether that happened is a different issue.'–(R026, Development partner)*

As further noted by **R026, Development partner**, '. . .the transport element was not factored in' at the formulation hence mothers would spend money on transport. At the formulation stage, the envisioned reimbursement of the scheme was as shown in Fig 2.

## Roles of the actors

**During formulation.** *A committee of actors was set up to discuss the policy formulation agenda*. The key informants noted that at the formulation of the current policy, the MoH set up a committee bringing together a mix of actors (such as development partners, the MoH representatives, and NHIF officials) which developed a concept note (comprehensive implementation document on how Linda Mama would look like) to review the whole FM policy process and share tasks.

*Different actors played different roles in the policy formulation stage.* Obtained from document reviews and IDIs, *Tables 3* and *5* summarises the actors who participated in the policy formulation and their roles. Characteristically, the influential participants at the formulation stage were the development partners (The World Bank, WHO, JICA, UNFPA, and USAID), who all supported the initial technical design. Of all the partners, the World Bank and JICA were most involved, as they were the co-founders of the previous FM policy. At the national level, the Presidency outlined the agenda; the appointed officials at the National Treasury allocated the budget; and the MoH, through the Principal Secretary (PS) and the director general's (DG) office provided oversight. Interestingly, while FM policy was targeted at improving maternal and reproductive health, the members of the MoH reproductive health department felt that they were not adequately involved as noted by some respondents:

> *'but not the team from the reproductive health stakeholders, they have not been largely involved, I mean, that is why I was saying we are all in the dark like facilities.'–(R023, MoH Official).*

Equally, despite the policy aiming at improving the QoC provision of maternal care, the MoH players managing the quality and standards felt they were not adequately included as noted by some respondents:

> *'that the players who are charged with quality just need to be roped in.'–(R034, MoH Official)*

The other players involved included the Council of Governors (CoG), which provided the modalities of implementation at the counties; the NHIF as the chosen purchaser of the fund, which provided implementation framework; the member of religious interest groups, private sector alliance interest group, and workers unions. Population Services International (PSI) was one of the international NGOs included in the process, which focused on persuading the committee to empanel the private sector facilities especially those they worked with to improve access in the informal settlements.

From the document review, the civil society and the beneficiaries (individual citizens and HCWs from both the private and the public sector) were involved through community forums. However, there was a feeling that the mothers or their representatives, some county officials and the healthcare workers may not have been involved and did not know about the policy:

> *'Involved? You know now, okay the change I would talk about is maybe they start involving us the people on the ground'–(R014, Nursing Officer)*

> *'I: Were you involved in the design of the free maternity of the Linda Mama? R: No'–(R016, County level manager)*

**During implementation.**    *There was a different organizational arrangement and role of actors in implementation.* The analysis showed that implementation of the FM policy took a top-down approach in three levels: the national, county, and facility levels (Fig 3). There were more actors in the implementation than in the formulation.

*Central (national) government coordination in the arrangement.* The policy imperatives emerging from the national level were de facto priorities of the government as captured in the

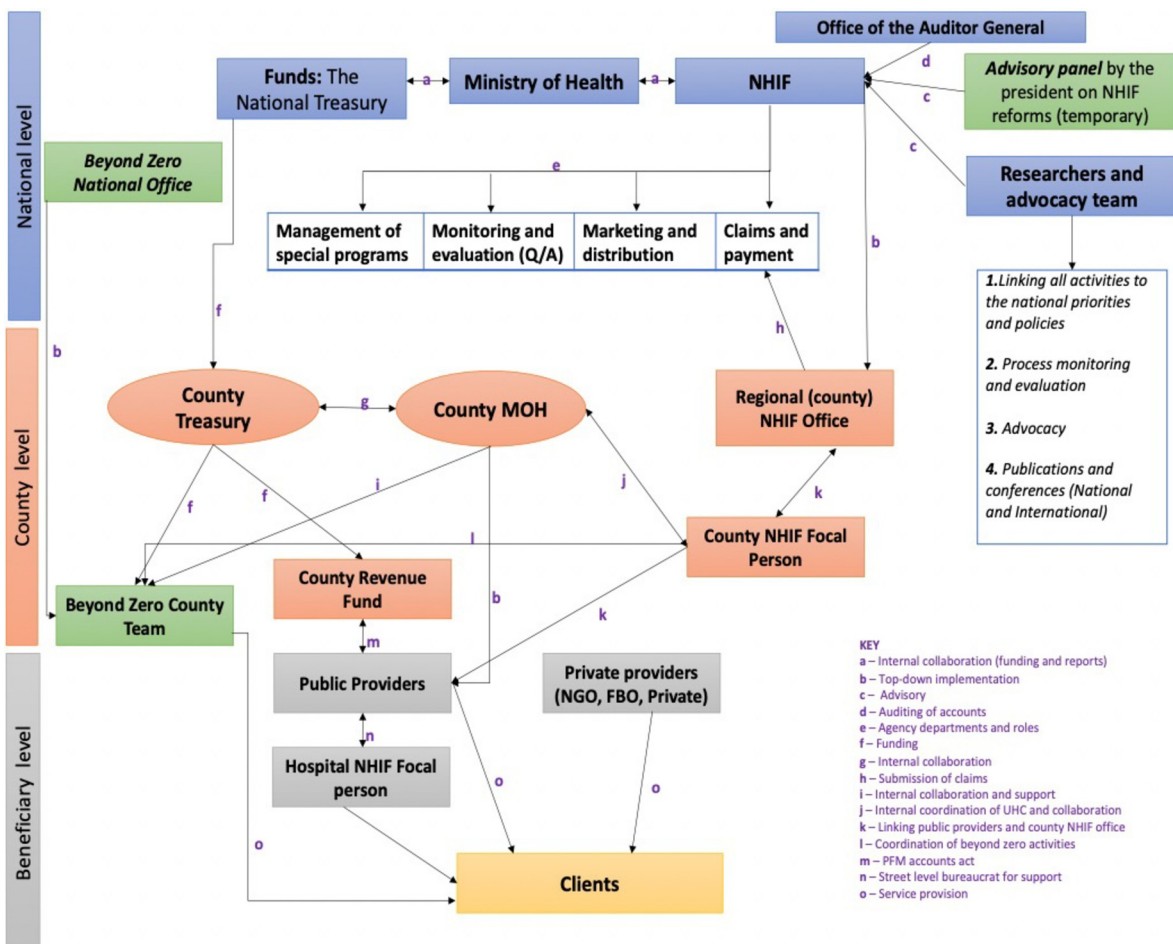

**Fig 3. Implementation arrangement of the free maternity policy as it is being implemented (source: Document review and interviews).**

president's *2017 Jubilee campaign Manifesto* [63]. At the national level, The Presidency, development partners, NHIF, MoH, The National Treasury, Office of the Auditor General, the CoG and the Parliament were joined up in what Exworthy and Powel [68] call 'horizontal dimension–joined-up government at the centre' and performed a multiple implementation roles as shown in *Tables 3* and *5*. MoH was a powerful and influential actor at the national level. Three entities in MoH: Cabinet Secretary (CS) Health, PS health, and DG Health, were strategic policy experts, who sourced funds from the National Treasury and provided strategic, future policy direction in line with the presidential directive of UHC (*Table 4*).

In terms of a governance structure, it was shown that it was imperative to have a proper reporting structure at the national level that would monitor the implementation of the program. However, there was a breakdown in the reporting channels that led to a gap at the national level. The effect was that the PS, who represented the MoH, would be receiving many communications concerning challenges of implementation from several sources which sometimes may have been incorrect. At MoH, the implementation was overseen by the equally powerful and influential department of preventative and promotive health (Division of Family Health), together with the NHIF, they have provided adequate social marketing to the policy through social mobilisation and communication of the providers and beneficiaries. Also, upon

receiving claims and utilisation reports from the NHIF, the MoH was able to track the level of remaining funds in the pot and mobilises additional reports from the National treasury. However, the team from division of family health at MoH–despite being concerned with the reproductive health–was not involved in the formulation but passively participate in the implementation as noted by one respondent:

'. . .but not the team from the reproductive health stakeholders, they have not been largely involved'–**(R023, MoH Official).**

Other departments at the MoH that were less powerful and had a medium level of interest were the division of health policy and planning, division of healthcare financing, standards and quality assurance and regulations, and monitoring and evaluation unit. The units provided strategic policy direction to the NHIF on their areas of strength and concern.

Similarly, the national treasury played a critical role but had a less influential role in the implementation process of the policy; however, they liaised with the parliament to approve the required budget for the policy. Since the introduction of the policy, they were able to provide the funds as required. Besides linking with the MOH, the National Treasury linked with the county treasury to provide other statutory funds not necessarily linked to the running of FM policy.

Equally, the NHIF was a powerful actor drawing from its mandate as an overall overseer of implementation of FM policy–as a managed fund under its department of programs and schemes–and primary purchaser of services. As a purchaser, the NHIF used its extensive network with service providers to accredit and contract providers–not previously registered on their system–for FM policy services provision. The NHIF timely reimbursed the providers for services rendered through its automated database for registration and authentication of beneficiaries. Using government ID numbers, NHIF verified the claims and redressed any complains arising from the providers on the mothers served. The NHIF had the mandate to report the claims and utilisation data for service provided, which was then yearly audited by the influential office of the auditor general. Besides, auditing the reports, the office of the auditor general was not concerned with the daily running of the implementation process.

The development partners, equally played critical roles role in the implementation such as developing financing strategies, demand generation, capacity building, and collaborations as noted by the respondents:

'We do capacity building at the provider level. . .. it's important that both the public and private understand the process of claim because of accreditation, process of contracting, understand issues to do with strategic purchasing in terms of . . .. service, how do they pay, how do they select just the whole aspect. Demand generation is one of our key aspects in terms of creation awareness.'–**(R029, Development partner)**

'We were working with NHIF to help them first of all package their informal sector product'–**(R030, Development partner)**

Still, at the national level, there were two other key players: the advisory panel, researchers and advocacy team, that were less influential in the process of implementation of the policy but played an important advisory role. The advisory panel was developed by the minister of health, albeit late in the implementation process (on 18th April 2019) in line with the Health Act of reforming and repositioning the NHIF as a strategic purchaser [69]. The team comprised the development partners, private sector, researchers, government technocrats, and

advocacy coalition teams and their role was to provide the technical and financial support for NHIF, part of which is management and implementation of the NHIF. On the other hand, the researchers, mostly research institutions, and the advocacy teams, mostly the civil society, were working independently or together with the NHIF to link the activities at the county and facility levels to the national priorities, participated in the process monitoring and evaluation, provided advocacy especially of the weak and vulnerable such as adolescents, and scientific publication which were meant to improve knowledge.

*Peripheral (county and health facility) coordination in the arrangement.* Further in the arrangement, there was joined-up governance at the periphery, where at the county level, several players worked towards the implementation of *Linda Mama*, as noted by one respondent: '*. . .it's almost everyone, it's like a teamwork*'–**(R004, Nursing Officer)**. The two key ministries at the county that played the biggest roles in the implementation process are the treasury and health. The county treasury was concerned with receiving finance from the national treasury and providing financial support and monitoring the flow of funds at the County Revenue Fund (CRF). On the other hand, the members of the MoH at the county who oversaw the implementation of the policy form the County Health Management Team (CHMT) and were composed of several dockets such as nursing, clinical services, monitoring and evaluation, research and development, pharmacy and administration. The dockets report to the county executive officer (CEC) of Health. Overall, the county governor [oversaw] *all the activities in the county, like especially in such free maternal its working and supervision.*'–**(R009, Nursing Officer)**.

However, as part of the CHMT, the most influential and active player in the implementation of the *Linda Mama* policy at the county was the chief officers of health as noted by the respondents:

'*So now our relationship to the county is probably linked to the chief officer, through the chief officer. Because if we have issues with the implementation, then we are supposed to address them to the chief. But other people we don't know because we don't really see them.*'–**(R005, Facility incharge)**

The county adopted the UHC agenda of the central government by employing a county focal person for NHIF, who was important but less influential player and had a role in *'moving forward not just with NHIF but the UHC goals of Kiambu County as a whole'*–**(R017, County level manager)**. The regional offices of the NHIF at the county also played a significant role of receiving, batching, and quality assurance check of all the claims from the facilities in the county and sending to the national offices.

At the service provider level, there were two kinds of providers: the private and the public providers, who provided services that were responsive to needs of clients and in line with contracted terms. The public providers were part of the previous FMS that was run before while the private sector joined the service in 2017 when the new service was moved to the NHIF. They all provided the service delivery as per the benefit package and reporting of services:

"*I will say that the hierarchy and the organogram of hospital management kicks into play any time there is an issue that touches on the hospital, whether it's Linda Mama or any other thing. We don't have separated organs to deal with Linda Mama outside other operational issues.*"–**(R010, Facility incharge)**.

Finally, the most interested but less powerful actors were the beneficiaries. They were responsible for registering with the NHIF either through self-registration or HCWs assisted,

utilised services and provided feedback. A summary of all roles, interests and power are in *Tables 3–5*.

## Discussion

This study examined the expanded FMP's background and context, the policy formulation processes, the policy content, and the actors' roles in formulation and implementation. One finding on the background and the context of the expanded policy was that it was mainly a political initiative driven by campaign promises and the need to align the functions with the national legal frameworks. It aimed to achieve international (SDGs) and national goals (UHC and improved access to SBA), addressing shortcomings from the previous 2013 FMP to enhance its effectiveness. These policy priorities and objectives converged at the right time (during an election year, when the previous policy had just been evaluated and needed change, and the government was charting a path towards achieving UHC), creating a 'window of opportunity' [as described by Kingdon [70]] to fully develop a robust FMP policy. The confluence incorporated the political importance of agenda-setting in policy reforms, as emphasised by Gilson et al. [71]; the necessity of prioritising policy agendas to align with national and international goals, demonstrated by Meessen et al. [72]; and the continuity of building upon the existing FMP within the policy agenda. Moreover, by learning from the previous FMP, it adopted a dynamic approach rather than remaining static. The finding shows that the policy formulation involved 'mixed scanning and/or muddling through,' [19] which means the formulation decision-making was flexible by balancing strategic FMP changes with addressing immediate policy needs and taking a gradualist approach to navigating existing constraints and political realities. This finding mirrors the practices of other countries. For instance, in Nepal, the converging interests–political and others–predestined the policy as an ideal vehicle for meeting the fortunes and objectives of the maternal incentive scheme [73].

Our finding on the policy formulation processes indicated that involving the private sector in design discussions was crucial, given their significant role in health service provision and their influence in policy decision-making. The private sector's role in healthcare delivery is significant and growing, particularly in sub-Saharan Africa, with for-profit entities providing 35% of outpatient care and informal providers contributing an additional 17% [74]. From our results, private sector leveraged its power and inclusion to implement the policy based on its strengths of having previously developed systems such as an enhanced network of hospitals and community health volunteers and accreditation and quality monitoring standards and guidelines mirroring those of the implementing body NHIF. Studies have shown that such network strengths fostered the public-private relationship, thereby increasing private provider accreditation into the health systems and a collegial relationship that had given small private providers more voice in the health system and improved health outcomes [75,76].

Some interests, particularly the setting of prices for new private sector entrants in the policy, were contentious, with private sector representatives justifying higher reimbursement rates. The network of private sector devised methods to bypass the political process at the formulation to engage the sector implementation players (network facilities) and shape the price debate at the grassroots level rather than at the top. They were what Sabatier and colleagues [77–79] label as policy advocates who dominated sub-policy coalitions of actors/stakeholders. Similar strategies have been used by organisations like the Africa Health Market for Equity (AHME) program, where social franchising improved Linda Mama's policy performance, with 79% of social franchising facilities participating [80]. However, their contributions depended on governance prerequisites, including institutions, management capacities, and collaborative culture for effective partnerships and targeted delivery designs [81].

The findings indicate that the expanded policy design aimed to enhance coverage and administrative efficiency, aligning with other studies [67]. The decision to make NHIF the ultimate purchaser of services was intended to improve the policy's sustainability, ease reimbursement logistics, and ease any legal hurdles [66,82]. While costing was done with appropriate assumptions and support from development partners, some key MoH players felt excluded, indicating a need for more collaboration. The costed benefits package aimed to allocate resources adequately and compensate for additional costs women incurred, given the government's subsidy for public health services [83]. However, the private sector, NGOs, and FBOs found the proposed reimbursements unattractive due to their high investments in infrastructure and staff. Studies show government facilities have lower costs per service unit than FBOs, NGOs, and private entities, with significant cost variability in private outpatient services, suggesting differences in productivity or quality [84].

The findings reveal that the policy design, marked by conflicts and time pressures, required a collaborative approach to find common ground for design and costing differences. Despite differing interests, deliberations and dialogue were essential for leadership and conflict management. This approach led to the preparation of key policy documents, including a Cabinet Memorandum for increased allocation to free maternity services, implementation guidelines, a technical policy proposal, and a communication strategy [82].

Regarding the actors' roles in formulation, our study indicates that the actors established an interaction committee to freely discuss and debate the formulation design and agenda. Studies show such collaborative approaches enable relevant players to devise novel solutions, fostering joint commitment and responsibility for implementation [18]. This collaboration facilitates a collective exploration of policy problems, emphasizing their urgency and solvability. Dye's [85] notes that public policies often reflect the interests of governing elites, which was evident in the committee. Although the Presidency was not active in the committee, technocrats had to align the design with the political agenda of achieving UHC. Our findings also reveal that participation in policy formulation was not fully inclusive. As noted by Grindle and Thomas [86]), despite the personal attributes, loyalties, institutional and political commitments, and training of policy actors, they are never completely autonomous. These actors had to navigate a complex context, addressing the problems and issues they faced, and provide solutions that were economically, politically, and administratively feasible. These findings align with existing literature on FM policy formulation [87], highlighting that decision-making within the FMP was shaped by a complex interplay of factors. It highlights the complexity of policy development and the persistent challenges in capturing diverse actors' resources, values, beliefs, and power dynamics [27,88].

Further, the appointed government officials (often the technocrats in government), the development partners, and the representatives significantly influenced the details of the policy formulation. While beneficiary representatives were present, the study revealed they were unaware of public participation opportunities. The level of public involvement in reforms underscores Grindle and Thomas's [86] distinction between bureaucratic compliance reforms with limited or 'invisible' public participation and those requiring visible engagement or comprehensive public involvement. The finding shows that the beneficiaries' representatives played a crucial role as agenda advocates, echoing participant interests. This findings align with the Ghana's FM policy formulation where the participants were categorised as agenda directors, approvers, advisors, and advocates, highlighting the active role of advocates, including beneficiaries, throughout the process [20]. Overall, while policy processes were well coordinated at the formulation stage, the inadequate involvement of beneficiaries and implementers raises questions about their commitment to implementing decisions in which they had limited input.

At the implementation level, more actors at both national and county levels supported achieving audit, research, financing, and strategic operational goals crucial for implementing the policy. These roles affirmed other research emphasising the mobilisation of specific capacities to address policy challenges and support implementation [89]. While implementers understood their roles as formulated, communication complexities at the national level arose from unclear departmental roles, leading to competing demands on the influential PS Health for strategic decisions. Interdependencies among units slowed implementation progress, although three (clear objectives, political support, and resources) of Hogwood and Gunn's proposed ten principles (clear objectives, political support, resources, coordination, management structure, monitoring and evaluation, flexibility, communication, public support, and learning and adaptation) for successful implementation were achieved [90]. At the county and facility levels, better interaction and coordination of roles have been observed, with counties hiring coordinators to facilitate communication across departments. Our findings underscore the importance of implementation readiness, ensuring that responsible organisations and stakeholders accept and support policy legitimacy amid sustained political backing and clear local context alignment [91–93].

Our study has some limitations. Firstly, since we evaluated implementation components in a single county out of Kenya's 47 counties, the results may not be generalisable due to the counties' heterogeneity. However, the identified contextual factors can enhance the transferability of findings to other counties, aiding in interpreting implications in diverse settings. Secondly, our study did not include local actors such as chiefs and community leaders, who are crucial in sub-Saharan Africa, representing a limitation in data collection.

## Conclusion

Applying a conceptual framework focusing on policy background, context, formulation processes, content, and the roles of actors in both formulation and implementation was valuable in explaining the political process that led to the expanded FMP policy. The formulation and agenda-setting of the expanded FMP illustrate how various agendas, such as political imperatives and global and national goals, drove policy change while building upon previous policy learnings. Actor power significantly influenced the policy's direction, reflecting their interests, ideas, and capacity to drive agenda decisions. For instance, the private sector leveraged its power and inclusion to implement the policy based on its strengths, such as a network of hospitals, community health volunteers, and accreditation and quality monitoring standards akin to NHIF. The expanded policy aimed to enhance coverage and administrative efficiency, with NHIF becoming the ultimate purchaser of services to improve sustainability, ease reimbursement logistics, and address legal hurdles. The policy design, marked by conflicts and time pressures, required a collaborative approach to reconcile design and costing differences. Despite differing interests, deliberations and dialogue were essential for leadership and conflict management, leading to the preparation of key policy documents. Actors established an interaction committee to freely discuss and debate the formulation design and agenda. Collaborative approaches enabled relevant players to devise novel solutions, fostering joint commitment and responsibility for implementation. Government officials, development partners, and representatives significantly influenced policy formulation details. While beneficiary representatives were present, they were unaware of public participation opportunities. At the implementation level, national and county-level actors supported achieving audit, research, financing, and strategic operational goals crucial for policy implementation. This study highlighted that decision-making within the expanded FMP was shaped by a complex interplay of contextual factors, processes, content, actors, interests, power dynamics, and roles. It underscores the enduring

relevance of policy analysis frameworks and theories, acknowledging the intricate nature of policy development. These insights can benefit other countries designing or redesigning FMP policies or other healthcare priority-setting processes and offer useful insights to local and international academic communities on the applicability of policy analysis theories in examining political processes for formulating healthcare policies.

## Supporting information

**S1 Checklist. Inclusivity in global research questionnaire.**
(DOCX)

**S1 Appendix. Interview guides.**
(PDF)

## Acknowledgments

The authors acknowledge all those who participated in the study and donated their time and skills.

## Author Contributions

**Conceptualization:** Boniface Oyugi, Sally Kendall, Stephen Peckham.

**Data curation:** Boniface Oyugi, Zilper Audi-Poquillon.

**Formal analysis:** Boniface Oyugi, Zilper Audi-Poquillon.

**Funding acquisition:** Boniface Oyugi, Zilper Audi-Poquillon.

**Investigation:** Boniface Oyugi.

**Methodology:** Boniface Oyugi, Zilper Audi-Poquillon.

**Project administration:** Boniface Oyugi.

**Resources:** Boniface Oyugi.

**Software:** Boniface Oyugi.

**Supervision:** Sally Kendall, Stephen Peckham, Edwine Barasa.

**Validation:** Boniface Oyugi.

**Visualization:** Boniface Oyugi.

**Writing – original draft:** Boniface Oyugi, Zilper Audi-Poquillon.

**Writing – review & editing:** Boniface Oyugi, Zilper Audi-Poquillon, Sally Kendall, Stephen Peckham, Edwine Barasa.

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
