## [Decision Letter · Decision Letter 0]

29 Apr 2024

PGPH-D-23-02417

The policy formulation process, and the role of actors in the policy formulation and implementation process: A policy analysis of the Kenyan free maternity policy

Dear Dr. Authors,

Thank you for submitting your manuscript to PLOS Global Public Health. After careful consideration, we feel that it has merit but does not fully meet PLOS Global Public Health’s publication criteria as it currently stands. Therefore, we invite you to submit a revised version of the manuscript that addresses the points raised during the review process.

We look forward to receiving your revised manuscript.

Kind regards,

Genevieve Cecilia Aryeetey, Ph.D

Academic Editor

Journal Requirements:

2. Please provide separate figure files in .tif or .eps format only and remove any figures embedded in your manuscript file. Please also ensure all files are under our size limit of 10MB.

Additional Editor Comments (if provided):

Reviewers' comments:

Reviewer's Responses to Questions

**Comments to the Author**

1. Does this manuscript meet PLOS Global Public Health’s publication criteria? Is the manuscript technically sound, and do the data support the conclusions? The manuscript must describe methodologically and ethically rigorous research with conclusions that are appropriately drawn based on the data presented.

Reviewer #1: Partly

Reviewer #2: Yes

2. Has the statistical analysis been performed appropriately and rigorously?

Reviewer #1: N/A

Reviewer #2: N/A

3. Have the authors made all data underlying the findings in their manuscript fully available (please refer to the Data Availability Statement at the start of the manuscript PDF file)?

Reviewer #1: Yes

Reviewer #2: Yes

4. Is the manuscript presented in an intelligible fashion and written in standard English?

Reviewer #1: Yes

Reviewer #2: Yes

5. Review Comments to the Author

Reviewer #1: Summary: This study was based on the topic “The policy formulation process, and the role of actors in the policy formulation and implementation process: A policy analysis of the Kenyan free maternity policy”. The subject is very important, particularly for low-and-middle-income countries.

There are major and minor revisions present in the current submission that need to be addressed. Since this is a very important topic, I would like to strongly encourage the authors to consider addressing the issues raised to strengthen the work for reconsideration.

Abstract

Major Compulsory Revisions

Overall: The abstract in its current form needs more strengthening. The results should state clearly the contextual factors that led to the policy formulation. What policy processes were adopted. Are there any identified gaps in the policy content. Are there some powerful actors who were excluded but who you think would have made the policy better if they were included? Did the actors’ practices of power influence the policy development and implementation positively or negatively? So basically, what lessons can be learnt from the formulation and implementation of ‘Linda Mama” policy

Line 42: The authors mentioned they conducted an exploratory qualitative study which involved document review, key informant interviews (KIIs) with national stakeholders, and in-depth interviews. They however did not state the study design used. This needs to be provided

Line 44-46: The authors stated that the used a theoretical framework capturing the preliminary situation analysis of the policy, the processes, the content, and the stakeholders' roles in the formulation and implementation. It would be useful to state the specific framework that was used upfront to help the reader. Also, theoretical framework wasn’t mentioned in the manuscript but conceptual and analytic framework. These are different from theoretical framework. This has to be reconciled.

Line 50: SDGs should be written in full first. Also, the sentence ‘…to improve the quality of maternal and neonatal care and eliminate financial barriers’ seems incomplete. Eliminate financial barriers to what?

Body of Manuscript

Introduction

Minor Essential Revisions

Line 75: ‘due inadequate’ should be ‘due to inadequate’.

Line 78-79: the sentence “It provided a package of essential health services for pregnant women accessed by all in the targeted population based on need and not the ability to pay” is not clear

Line 85: the authors should consider replacing “researchers” which runs through the rest of the manuscript with “studies”.

Line 88: The word policy should be deleted since it’s more like a repetition from line 86.

Methods

Major Compulsory Revisions

The guiding conceptual and analytical framework

The health policy analysis triangle framework by Walt and Gilson is appropriate and adequate enough for this study. The framework captures the context, processes, content and actors involved in the formulation as well as the implementation of the policy and thus helps to better understand the health policy reform and to plan for effective implementation. I therefore think the use of Hercot et al and SHIELD project which are not to support the framework but as separate frameworks/applied approaches are redundant and should be deleted

The authors should briefly describe the health policy analysis triangle framework and state what constitutes the context, process, content and actor components based on this study and how they applied it in this study. What were assessed or examined under each component?

The statement “It captures the background of the policy, derived from Hercot et al.’s [25] work, which focuses on the preliminary situation analysis and setting the priorities of the policy (an understanding of the origin of the policy). It also emphasizes the portrayal of an existing window of opportunity needed to restrict the ‘inventory phase’ of a policy” is captured by the context of the policy process under the health policy analysis triangle framework when the framework is applied well and therefore should be deleted

The use of preliminary situation analysis should be replaced with context of the policy development

The statement “(whose roles, power, and influence during formulation and implementation was analysed through a stakeholder’s analysis [28-30])” should be deleted

Study design

Line 137: the authors should state the specific study design used, e.g. case study, cross-sectional etc

Study population and sampling

Line 152: FMP should be written in full first

Line 153-154: National health insurance agency does not correspond to NHIF

Data collection and analysis

The study lacks detailed description of the data collection procedure. Which data were collected under the context, process, content and actor components of the framework? What information were obtained from the informant categories. How was the data collected for the document review or what instruments were used. Which aspects of the policy development and implementation were covered by the instruments?

Line 167: BO needs to be defined

Results

Major Compulsory Revisions

This section requires reformatting. This is essential because the current format is very difficult to make sense of as the manner of presentation is difficult for the reader to make sense of the value to the research findings.

In particular, the presentation of headings/themes without sub-headings or subthemes. This makes it very difficult for the reader to appreciate the value of the research findings. I will recommend that the authors present the findings under five headings/themes: 1) context of FM policy formulation 2) the FM policy formulation processes 3) content of the FM policy 4) Influence of actors over FM policy formulation and 5) influence of actors over FM policy implementation. Then under each, have sub-themes based on inductive analysis instead of the current format.

Line 189-190: Authors should replace the heading with ‘context of FM policy formulation’

Line 193: FM should be written in full first

Line 195-196: Instead of bolding, a sub-heading/theme should be created.

Line 198: “the Vision 2030” of what?

Line 200:” with” consider delete

Line 208: a subtheme/heading is needed

Line 209: Which of the SDGs? There is the need to be specific

Line 218-219: a sub-theme/heading needed

Line 234: a subtheme/heading needed

Line 234-236: the sentence Some respondents perceived the FMS policy to be a political tool used by the government to fulfil a campaign agenda, as captured in the president’s Jubilee Party 2013 Presidential campaign Manifesto” is a bit confusing. The authors should make it clear whether it is the Linda Mama policy they evaluated or previous FM policy in 2013

Lines 236-237: The sentence “some of the respondents highlighted that the goal of the policy was to fulfil part of the Big Four agenda that the then…”. What are the Big Four agenda?

Line 246: a subtheme/heading needed

Lines 261-262: a subtheme/heading needed

Line 277-278: a subtheme/heading needed

Line 287: a subtheme/heading needed

Lines 290-291: Repetition of process evaluation. Consider delete

Line 292: what is inflated utilization number

Line 305: Authors should consider making the heading “the FM policy formulation processes’

Line 306: a subtheme/heading needed

Line 342: a subtheme/heading needed

Line 353 and 355: what are quantiles 1,2 and 3

Line 376: a subtheme/heading needed

Line 397: a subtheme/heading needed

Line 430-431: a subtheme/heading needed

Line 457: a subtheme/heading needed

Line 481-482: consider breaking this into two main headings: 1) Influence of actors over FM policy formulation and 2) influence of actors over FM policy implementation. What are the bottlenecks or successes identified?

Line 483: a subtheme/heading needed

Line 488: a subtheme/heading needed

Line 522: a subtheme/heading needed

Line 531: a subtheme/heading needed. Also, the use of “joined-up government at the centre”, what does it mean?

Line 590: “important role” in what?

Line 600: a subtheme/heading needed

Discussions

Major Essential Revisions

Overall, the discussion needs a major revision. The discussion should focus on the successes, failures/bottlenecks/constraints or the lessons learnt from the policy development and implementation. That is, it should speak to the findings on the contextual factors that led to the policy formulation, the policy processes adopted, any identified gaps in the policy content and the influence of the actors in formulation and implementation in the context of current literature.

Line 657: what window of opportunity did you find in this study

Line 662-663: What does the statement “Further, by building on the lessons from the previous FM policy, the policy is taking the ebbs and flows fluid process instead of remaining in a fixed static form” mean?

Line 668-669: What does “Sitting at the centre of the policy triangle is the issue of power, and its role in decision-making is incontrovertible” mean?

Line 677: “(such as the developed concept of social franchise networks)”. What does it mean and where is it coming from?

Line 683-685: “The NGOs representing the interests of the private sector in the informal settlements devised methods to bypass the political process at the formulation to engage the sector and shape the price debate at the grassroots level rather than at the top”. Is this found in the study?

Line 711: Faith Based Organisations should be written in abbreviated form

Line 712: Nongovernmental Organisations should be written in abbreviated form

Line 726: The use of “stakeholders” should be replaced with “actors”, its more appropriate. This should reflect in the relevant sections of the manuscript”

Line 750-751: This sentence “The finding shows that the beneficiaries’ representatives were significant because they were classified in the FM reform as the latter” is not clear

Line 766: What does PS mean

Conclusion

The conclusion should speak to the successes, bottlenecks/constraints or the lessons learnt from the policy development and implementation that can offer support for policy reform. That is, it should speak to the contextual factors that led to the policy formulation, the policy processes adopted, any identified gaps in the policy content and the influence of the actors in formulation and implementation.

Reviewer #2: General Comments

• The paper will benefit from few typos and grammar errors e.g. “due inadequate preparation before its rollout...”

• A section on limitation is missing. This should either come out clearly in the discussion section or have a stand-alone section.

Title

• Very relevant and useful to literature and policy framing particularly in developing country settings.

• However, the manuscript title is a little bit winding. I suggest that the title could just read “The role of actors in the policy formulation and implementation process: A policy of Kenya Free Maternal Health Policy.”

Abstract

• Loaded and not straight forward e.g.

“We conducted an exploratory qualitative study, which involved document review, key informant interviews (KIIs) with national stakeholders, and indepth interviews with County officials and health care workers (HCWS). We used a theoretical framework capturing the preliminary situation analysis of the policy, the processes, the content, and the stakeholders' roles in the formulation and implementation. This study was conducted in three facilities (levels 3, 4, and 5) in Kiambu County in Kenya.”

• Three sentences as quoted above may be merged into two to read well and straightforward.

“Policy formulation or change requires the agents to work within the relevant context, stakeholder interests, power, ideas and framing of issues.”

• This last statement in the abstract as quoted above serves no purpose. Is a statement of purpose, conclusion, or recommendation – for me the statement is not properly placed and should be taken out.

Introduction

• Well written and captures salient aspects.

• However, a few re-structuring will create a good flow. For example, line 108 to 116 states, “Kenya has made some good progress towards reducing neonatal and mortality rates and …. Kenya has made some good progress towards reducing neonatal and mortality rates and” (lines 108 -116)

• This entire paragraph may be captured in the opening paragraph to provide a good context of the “free” policy success to lay bare the rationale for examining its formulation and implementation. The usefulness of the policy gets buried where it is currently located.

Methodology

The guiding conceptual and analytical framework

• Well grounded. However, references mainly cited published articles.

Design

• Appropriate

Setting

“Kiambu County was chosen because of the logistic feasibility of data collection and the sociodemographic characteristics, health indicators and population size…”

• Logistical challenges are not good enough scientific reason for choosing a study site. This MUST always be accompanied by a compelling need for the site to be chosen. This is conspicuously mission in the section. Although, the authors cited sources to readers, authors have a duty to state what in those sources makes a compelling case for this county to be chosen. The population demographic stated afterwards don’t justify the choice either.

•

Study population and selection.

• Wide enough yet left out community opinion leaders and perhaps pregnant women.

“…county department of health officials; facility-in-charge and HCWs and others.” (Line 151 to 155).

• Who are the others in the above sentence? If these are traditional authorities or anyone from the community, it should stand out and as they are very critical in public policy formulation process. Otherwise, it should appear in the manuscript as a limitation of data gathering. You later read other in Table 5 to refer to Beyond Zero, Jacaranda Health etc and get a little bit confusion. Define others clearly and state whether or not traditional/opinion leaders were part of the process.

Results

“Preliminary situation analysis, setting the priorities, and the context of the policy.”

• The word “preliminary” is a little bit confusion and makes it look it the authors are presenting a ‘preliminary’ situational analysis rather than the first step of the policy formulation process. This is also reflected in the discussion (This preliminary situation analysis shows that the adoption /formulation of this policy followed the opening of a ‘window of opportunity”) section (line 656) and makes it worse. My considered opinion is that the authors adapt the theme to read without the word ‘preliminary’. It perhaps will read well and clear.

• Critical quotes are mission from the opening results, yet the authors made references to same. “...for example, as noted by the respondents, the constitution provided every citizen with the right to quality health and life (including maternal and neonatal care)...” - Which respondent and where did they note it?

• Solitary quotes referred to as bases for themes/subtheme e.g. The need to align with the global goals of achieving sustainable development goals (SDGs). The SDGs are a critical component of every country’s health systema and policy directive and as such should appear in not just from one KII. This should copiously be referred to in the results section. It never appears adequately in the interview responses then this should be highlighted in the discussion section. Other themes such “…to achieve political agenda…” is ok to have solitary quotes as this is a common knowledge in LMICs political space.

Discussion

• “…Our findings show that the policy redesign envisaged expanding coverage and enhancing”.

• Why is the word policy redesign being introduced at this section? It is not clear to me.

• The actors leave out chiefs and community/opinion leaders who are rather critical in sub-Saharan Africa. Although, elected leaders are technically representatives of chiefs and opinion leaders, in policy formulation process are the FMS, their role is essential and should be stated as policy formulation process gap, if it was missing in the process. Alternatively, if the authors did not collect data on same, then this should be stated as limitation of data collection.

Conclusion

• The conclusion is rather a little off to me – the main aspects have it as,

“Our study highlights the basis for changing the previous policy to the current one driven by multiple agendas, including political determinants, the need to achieve global and national goals, and learnings from the previous policy. The interconnectedness of these drivers shaped the content and policy formulation processes”.

• I though the authors would rather conclude on bases of the framework upon which the study was grounded and state whether the process of formulation and implementation followed their chosen farmwork of study and why not.

• “Understanding policy processes is relevant across other LMICs, and we hope that this framework contributes to policy analysis and learning in Kenya and beyond”-

• This is similar to what is captured in the abstract ending. It serves little to no purpose to me. It can be taken out with losing anything relevant.

6. PLOS authors have the option to publish the peer review history of their article (what does this mean?). If published, this will include your full peer review and any attached files.

**Do you want your identity to be public for this peer review?** For information about this choice, including consent withdrawal, please see our Privacy Policy.

Reviewer #1: No

Reviewer #2: **Yes: **Dr John Azaare

---

## [Decision Letter · Decision Letter 1]

27 Sep 2024

Policy formulation and actor roles in the expanded Kenyan free maternity policy (Linda Mama): A policy analysis

PGPH-D-23-02417R1

DearAuthors,

We are pleased to inform you that your manuscript 'Policy formulation and actor roles in the expanded Kenyan free maternity policy (Linda Mama): A policy analysis' has been provisionally accepted for publication in PLOS Global Public Health.

Best regards,

Genevieve Cecilia Aryeetey, Ph.D

Academic Editor

None

Reviewer Comments (if any, and for reference):

Reviewer's Responses to Questions

**Comments to the Author**

1. If the authors have adequately addressed your comments raised in a previous round of review and you feel that this manuscript is now acceptable for publication, you may indicate that here to bypass the “Comments to the Author” section, enter your conflict of interest statement in the “Confidential to Editor” section, and submit your "Accept" recommendation.

Reviewer #1: All comments have been addressed

Reviewer #2: All comments have been addressed

2. Does this manuscript meet PLOS Global Public Health’s publication criteria? Is the manuscript technically sound, and do the data support the conclusions? The manuscript must describe methodologically and ethically rigorous research with conclusions that are appropriately drawn based on the data presented.

Reviewer #1: Yes

Reviewer #2: Yes

3. Has the statistical analysis been performed appropriately and rigorously?

Reviewer #1: N/A

Reviewer #2: N/A

4. Have the authors made all data underlying the findings in their manuscript fully available (please refer to the Data Availability Statement at the start of the manuscript PDF file)?

Reviewer #1: Yes

Reviewer #2: Yes

5. Is the manuscript presented in an intelligible fashion and written in standard English?

Reviewer #1: Yes

Reviewer #2: Yes

6. Review Comments to the Author

Reviewer #1: I have reviewed the revised manuscript and think all of the concerns I raised in the previous version have been satisfactorily addressed. Congratulations to the authors for the excellent work, which contributes to our understanding of health policy process, particularly in low-and-middle-income settings.

Reviewer #2: line 833 "our study has several limitations...." then the authors proceed to state two. I don't find two to be "several". The authors might want to revise the sentence.

7. PLOS authors have the option to publish the peer review history of their article (what does this mean?). If published, this will include your full peer review and any attached files.

**Do you want your identity to be public for this peer review?** For information about this choice, including consent withdrawal, please see our Privacy Policy.

Reviewer #1: **Yes: **Dominic Dormenyo Gadeka, MPH, PhD

Reviewer #2: No
